# Small-Molecule Anti-HIV-1 Agents Based on HIV-1 Capsid Proteins

**DOI:** 10.3390/biom11020208

**Published:** 2021-02-03

**Authors:** Takuya Kobayakawa, Masaru Yokoyama, Kohei Tsuji, Masayuki Fujino, Masaki Kurakami, Sayaka Boku, Miyuki Nakayama, Moemi Kaneko, Nami Ohashi, Osamu Kotani, Tsutomu Murakami, Hironori Sato, Hirokazu Tamamura

**Affiliations:** 1Institute of Biomaterials and Bioengineering, Tokyo Medical and Dental University, Chiyoda-ku, Tokyo 101-0062, Japan; tkobmr@tmd.ac.jp (T.K.); ktsuji.mr@tmd.ac.jp (K.T.); m-kurakami@hhc.eisai.co.jp (M.K.); ma190086@tmd.ac.jp (S.B.); ma190076@tmd.ac.jp (M.N.); kanekomo@nissanchem.co.jp (M.K.); ohashi@ac.shoyaku.ac.jp (N.O.); 2Pathogen Genomics Center, National Institute of Infectious Diseases, Musashimurayama, Tokyo 208-0011, Japan; yokoyama@nih.go.jp (M.Y.); konioo@nih.go.jp (O.K.); 3AIDS Research Center, National Institute of Infectious Diseases, Shinjuku-ku, Tokyo 162-8640, Japan; fmasa@niid.go.jp

**Keywords:** anti-HIV, capsid, dipeptide mimic, in silico screening

## Abstract

The capsid of human immunodeficiency virus type 1 (HIV-1) is a shell that encloses viral RNA and is highly conserved among many strains of the virus. It forms a conical structure by assembling oligomers of capsid (CA) proteins. CA dysfunction is expected to be an important target of suppression of HIV-1 replication, and it is important to understand a new mechanism that could lead to the CA dysfunction. A drug targeting CA however, has not been developed to date. Hydrophobic interactions between two CA molecules via Trp184/Met185 in CA were recently reported to be important for stabilization of the multimeric structure of CA. In the present study, a small molecule designed by in silico screening as a dipeptide mimic of Trp184 and Met185 in the interaction site, was synthesized and its significant anti-HIV-1 activity was confirmed. Structure activity relationship (SAR) studies of its derivatives were performed and provided results that are expected to be useful in the future design and development of novel anti-HIV agents targeting CA.

## 1. Introduction

As a retrovirus, human immunodeficiency virus type 1 (HIV-1) can infect CD4-positive T-cells or macrophage, eventually causing acquired immunodeficiency syndrome (AIDS). To date, many anti-HIV-1 drugs [1,2,3], such as inhibitors of reverse transcriptase [4], protease [5] and integrase [6,7], have been developed for the therapeutic treatment of HIV-1-infected individuals and AIDS patients. The utilization of the above drugs in combination with antiretroviral therapy (cART) has brought remarkable success to the chemotherapy of HIV infectious diseases [1,2,3]. There are serious defects however, which have not been escaped. These contain the appearance of mutant viral strains with multi-drug resistance, emergence of severe side effects and costs of the dosed drugs. In an effort to solve these problems and enhance the repertoire of anti-HIV-1 drugs, we have sought drugs with different mechanisms of action such as coreceptor CXCR4 antagonists [8,9,10,11,12,13,14], CD4 mimics [15,16,17,18,19], fusion inhibitors [20,21,22,23], integrase inhibitors [24,25,26] and inhibitors of viral uncoating and viral assembly [27,28,29,30].

HIV-1 capsid (CA) proteins [31,32], which are generated from the Gag precursor protein Pr55Gag and are composed of *N*- and *C*-terminal domains (NTD/CTD), are highly conserved among many HIV strains. These proteins are structurally assembled by oligomerization of hexamers and pentamers [33], to form a CA core with a conical structure [34,35], which encapsulates the HIV-1 RNA genome, the integrase and reverse transcriptase. Matrix (MA) proteins also result from Pr55Gag, located inside viral membranes, and contribute to the assembly of the virion shell [36,37]. Both the MA and CA proteins are considered to be great targets for inhibition against viral replication, and some MA- and CA-derived peptides with anti-HIV activity have been reported to date by our group [27,28,29,30] and others [38,39,40,41]. Since viral uncoating based on the MA/CA degradation and viral assembly as a consequence of MA/CA protein oligomerization are performed inside host cells, inhibitors must have cell membrane permeability to be able to suppress viral uncoating and assembly. Consequently, an octa-arginyl group [42] was incorporated into the above peptide inhibitors to add cell membrane permeability [27,28,29,30]. However, small compounds that are found to have inhibitory activity against viral uncoating and assembly might have cell membrane permeability. To date, several small compounds have been discovered [43,44,45,46,47,48,49,50,51,52,53,54,55,56,57,58,59] but, except for GS-6207 [57], none has progressed to clinical trials.

## 2. Materials and Methods

### 2.1. General Information

All reactions utilizing air- or moisture-sensitive reagents were performed in dried glassware under an atmosphere of nitrogen, using commercially supplied solvents and reagents unless otherwise noted. CH_2_Cl_2_ (DCM) was distilled from CaH_2_ and stored over molecular sieves 4A. Thin-layer chromatography (TLC) was performed on Merck 60F_254_ precoated silica gel plates (Merck, Darmstadt, Germany) and was visualized by fluorescence quenching under UV light and by staining with phosphomolybdic acid, p-anisaldehyde, or ninhydrin. A solvent system consisting of 0.1% TFA in H_2_O solution (*v*/*v*, solvent A) and 0.1% TFA in MeCN (*v*/*v*, solvent B) were used for HPLC elution. For analytical HPLC, a Cosmosil 5C_18_-ARII column (4.6 × 250 mm, Nacalai Tesque, Inc., Kyoto, Japan) was employed with a linear gradient of B at a flow rate of 1 cm^3^min^−1^ on a JASCO PU-2089 plus (JASCO Corporation, Ltd., Tokyo, Japan), and eluting products were detected by UV at 220 nm. Preparative HPLC was performed using a Cosmosil 5C_18_-AR II column (20 × 250 mm, Nacalai Tesque, Inc.) on a JASCO PU-2086 plus (JASCO Corporation, Ltd.) in a suitable gradient mode of B at a flow rate of 10 cm^3^min^−1^. Optical rotations were measured on a JASCO P-2200 polarimeter (JASCO Corporation, Ltd.) operating at the sodium D line with a 100 mm path length cell at 25 °C, and were reported as follows: [α]_D_ (concentration (g/100 mL), solvent). Infrared (IR) spectra were measured on a JASCO FT/IR 4100 (JASCO Corporation, Ltd.) and recorded as wavelength (cm^−1^). ^1^H- and ^13^C-NMR spectra were recorded using a Bruker AVANCE III 400 spectrometer or AVANCE 500 spectrometer (Bruker, Billerica, MA, USA). Chemical shifts are reported in δ (ppm) relative to Me_4_Si (in CDCl_3_, or MeOH-*d_4_*) as an internal standard. High-resolution mass spectra were recorded on a Bruker Daltonics micrOTOF focus (ESI) mass spectrometer (Bruker) in the positive detection mode. For flash chromatography, silica gel 60 N (Kanto Chemical Co., Inc., Tokyo, Japan) was employed.

The details of synthesis and characterization data of the compounds are available in the Appendix A.

### 2.2. In Silico Screening of Antiviral Candidates

To perform the in silico screening, we first obtained the structure of the dimer of CA proteins (PDB ID: 3J34) from the Protein Data Bank (https://www.rcsb.org/). The structure of the dimer of CA proteins was thermodynamically optimized by the energy minimization using MOE and the Amber10: EHT force field [60,61]. The one monomer of the dimer of CA proteins was fixed as a receptor, whereas the residues of the other monomer were removed except for Typ184 and Met185 residues, of which the side chains play a key role of the dimer formation. Using the complex composing of the CA monomer as a receptor and Trp184Met185 dipeptide as a ligand, we searched for the compounds having a higher affinity than the dimer formation. To do this, the main chain backbone replacement of the dipeptide on the CA protein was performed by the Scaffold Replacement application in MOE using the linker database of MOE and the Amber10: EHT force field, whereas the side chains of the dipeptide were fixed. From the result, we selected the compounds having higher scores of the binding affinity for receptor (London dG), ligand efficacy, and topological polar surface area (TPSA).

### 2.3. Evaluation of Anti-HIV-1 Activity and Cytotoxicity

For virus preparation, 293T/17 cells (Invitrogen), which are maintained in Dulbecco’s modified Eagle medium (DMEM) containing 10% FBS, in a T-75 flask were transfected with 10 μg of the pNL4-3 construct by the calcium phosphate method. The supernatant was collected 48 h after transfection, passed through a 0.45 μm filter, and stored at −80 °C as a stock virus. Inhibitory activities of test compounds against X4-HIV-1 (NL4-3 strain)-induced cytopathogenicity in MT-4 cells [62], which are maintained in RPMI-1640 containing 10% FBS, were assessed by the MTT assay. Various concentrations of test compound solutions were added to HIV-1-infected MT-4 cells at multiplicity of infection (MOI) of 0.001, and placed in wells of a 96-well microplate. The test compounds and the reference compounds such as AZT and AMD3100 were diluted by two-fold and five-fold, respectively. After 5 days’ incubation at 37 °C in a CO_2_ incubator, the number of viable cells was determined by the MTT assay. Cytotoxicities of the test compounds were determined based on reduction of the viability of MT-4 cells by the MTT assay. The p24 antigen content in the culture supernatant was measured using HIV-1 p24 antigen enzyme-linked immunosorbent assay (ELISA) kit according to manufacturer’s instructions (Zeptometrix, Buffalo, NY, USA). A reverse transcriptase inhibitor, AZT, a CXCR4 antagonist, AMD3100 (Sigma Aldrich, St. Louis, MO, USA) were employed as positive control compounds with anti-HIV activity.

## 3. Results

### 3.1. In Silico Screening to Find Drug Leads Targeting CA Proteins

#### 3.1.1. Structural Analysis of CA Proteins

Structural analysis of CA proteins [33,34,35,63] revealed a hydrophobic interaction between two CA molecules in Helix 9 of CTD involving Trp184 of one molecule and Met185 of the other molecule. This interaction is important for stabilization of the multimeric structure forming the CA core (Figure 1). In addition, viral mutants with Trp184Ala and Met185Ala mutations have no infectivity because the dimeric interaction between two CA molecules of the mutants is weakened, thereby causing abnormal morphology of the viral particles [64]. Furthermore, a CA-derived fragment peptide covering Helix 9, which includes Trp184 and Met185, was previously found to have significant anti-HIV activity [29], and the Trp184-Met185 dipeptide is known to be highly conserved among natural HIV/simian immunodeficiency virus (SIV) strains [65]. In this study, based on the above information, the site of the hydrophobic interaction between two CA molecules via Trp184 of one molecule and Met185 of the other molecule might be considered a valid drug target for CA dysfunction. A novel small molecule, which might bind to the above site, was designed by in silico screening. This drug candidate and several of its derivatives were synthesized, and their anti-HIV activity and cytotoxicity were evaluated.

#### 3.1.2. Results of the In Silico Screening

Initially, a series of dipeptide mimics of Trp184 and Met185 were designed using the Molecular Operating Environment (MOE) (Chemical Computing Group Inc., Montreal, QC, Canada). Briefly, using the structure of the CA protein dimer (PDB ID: 3J34), the structure of one monomer molecule was fixed as a receptor side, and the main chain of Trp184 and Met185 of the other monomer molecule was removed. The side chains of these two residues were fixed in place, and the backbone structures crosslinking two side-chain functional groups were screened against the linker database provided by MOE to bind to the receptor side (Figure 2). Antiviral candidates of dipeptide mimics were selected using the Scaffold Replacement application in MOE and based on the following scores showing binding affinity for receptors, ligand efficacy (London dG), topological polar surface area, molecular weight, log of the octanol/water partition coefficient (SlogP) and an estimate of the synthetic feasibility [66]. Many London dG values mean binding affinities of available compounds for target proteins, and smaller values show higher binding affinity. The London dG value of the dimer of CA proteins is approximately −6, and compounds with London dG values of less than −6 have higher binding affinity for a CA molecule when compared to the interaction between two CA molecules. This screening served to identify some candidates with useful binding affinity, including MKN-1 (**1**), whose London dG value, ligand efficiency, topological polar surface area (TPSA) and SlogP value are −9.134 kcal/mol, 0.2559, 69.73 Å^2^ and 4.022, respectively. The structure of MKN-1 (**1**) is completely different from that of known small compounds, which were previously developed [43,44,45,46,47,48,49,50,51,52,53,54,55,56,57,58,59].

### 3.2. Synthesis of Novel CA-Targeting Anti-HIV Agents

#### 3.2.1. Synthesis of MKN-1 (**1**)

A possible synthesis of MKN-1 (**1**) was outlined. For its construction, the structure of **1**, with two chiral centers was divided into three segments (Scheme 1).

Initially, Segment I was prepared as a mixture of racemates. Treatment of 1,3-butanediol (**2**) with *p*-toluenesulfonyl (tosyl) chloride (TsCl) in the presence of a catalyst, 4-dimethylaminopyridine (DMAP) gave the tosylated alcohol (**3**) [67], and subsequent treatment with sodium methanethiolate led to a sulfide (**4**) that corresponds to Segment I, in 96% yield over two steps (Scheme 2a). The synthesis of Segment II is shown in Scheme 2b. Treatment of the indole (**5**) with iodine in the presence of potassium hydroxide yielded the 3-iodinated indole (**6**), and subsequent *N*-Boc-protection gave a Boc-protected indole (**7**) in 60% yield over two steps. Treatment of **7** with *n*-butyllithium and isopropoxyboronic acid pinacol ester produced a pinacol ester (**8**) corresponding to Segment II, in 50% yield (Scheme 2b) [68]. Segment III was stereoselectively synthesized using a Strecker reaction [69]. Treatment of *o*-bromobenzaldehyde (**9**) with (*S*)-(-)-1-(4-methoxyphenyl)-ethyl amine (**10**) and sodium cyanide gave (*S*,*S*)-α-aminonitrile (**11**) in a highly diastereoselective reaction, in 56% yield. The acid hydrolysis of **11** led to an enantiopure (*S*)-α-arylglycine (**12**), and the subsequent methyl esterification with thionyl chloride and *N*-Boc-protection of the α-amino group in (**13**) produced compound **14** that corresponds to Segment III, in 69% yield after three steps (Scheme 2c).

Using these three segments, MKN-1 (**1**) and its diastereomer were synthesized as shown in Scheme 2d. Compounds **8** (Segment II) and **14** (Segment III) were condensed by a Suzuki-Miyaura cross coupling reaction using tetrakis(triphenylphosphine)palladium (0) to obtain compound **15** in 98% yield. Saponification of compound **15** with lithium hydroxide yielded an acid (**16**), and the subsequent condensation with compound **4** (Segment I) by 1-ethyl-3-(3-dimethylaminopropyl) carbo-diimide hydrochloride (EDCI·HCl) in the presence of a catalyst (DMAP) gave an ester (**17**). The deprotection of the two *N*-Boc groups of **17** by HCl/dioxane gave MKN-1 (**1**) and its diastereoisomer, which were separated and purified by preparative HPLC to yield a first eluent MKN-1A (**1A**) and a second eluent, MKN-1B (**1B**), in yields of 3% and 2%, respectively, over three steps (Scheme 2d).

#### 3.2.2. Stereoselective Synthesis of MKN-1 (**1**)

Next, the stereoselective synthesis of MKN-1 (**1**) was performed. In the synthesis of Segment I, (*S*)-(+)-1,3-butanediol (**18**) was used as a starting material, and a chiral alcohol (**20**) was obtained in 63% in two steps in a manner (Scheme 3) similar to that shown in Scheme 2a. Saponification of the ester (**15**), the subsequent condensation with an alcohol (**20**) and the *N*-Boc deprotection followed by HPLC purification obtained diastereoselectively the target compound MKN-1 (**1**) in 12% yield over three steps in a pathway similar to that shown in Scheme 2d (Scheme 3). In HPLC analysis, MKN-1 (**1**) corresponds to MKN-1A (**1A**) (Appendix A).

#### 3.2.3. Synthesis of MKN-1 Derivatives

##### Synthesis of MKN-1 Derivatives with Aryl Ring Substitution **22**, **24**, **27**, and **28**

MKN-1 (**1**) has two pharmacophoric functional groups: the indolyl and sulfidyl groups. Initially, MKN-1 derivatives, in which the indolyl moiety has been replaced, were designed. In general, naphthyl groups and benzofuranyl and benzothiophenyl groups are used as useful analogues of an indolyl group. Thus, MKN-1 derivatives, with naphthyl, benzofuranyl and benzothiophenyl groups in the place of the indolyl group, were synthesized. In the synthesis of MKN-1 derivatives with a 1-naphthyl or 2-naphthyl group, the Suzuki-Miyaura cross coupling of a phenylglycine derivative (**14**) with 1-naphthylboronic acid or 2-naphthylboronic acid, led to compound **21** or **23**, both in 42% yield (Scheme 4a,b). Subsequent saponification, condensation with compound **20**, deprotection of the *N*-Boc group and HPLC purification gave compound **22** in 12% yield or **24** in 10% yield, over a three-step route similar to Scheme 2 and Scheme 3 (Scheme 4a,b). In the synthesis of MKN-1 derivatives with a benzofuranyl or benzothiophenyl group, the Suzuki-Miyaura cross coupling of a phenylglycine derivative (**14**) with benzofuran-3-boronic acid or benzo[*b*]thiophene-3-boronic acid followed by the treatment described above gave compound **27** in 5% yield or **28** in 2% yield, in four steps (Scheme 4c).

##### Synthesis of MKN-1 Derivatives with Sulfide Substitution **30**, **34**–**36**, and **38**

Next, MKN-1 derivatives, in which the sulfidyl moiety was replaced, were designed. MKN-1 derivatives, with methoxy, *tert*-butyl sulfidyl, *iso*-propyl sulfidyl, benzenesulfidyl and methanesulfonyl groups in the place of the methanesulfidyl group, were synthesized. In the synthesis of an MKN-1 derivative with a methoxy group, the alcohol bearing a methoxy group (**29**) was prepared by methylation of compound **18** in 17% yield, and saponification of compound **15**. Subsequent condensation with alcohol **29**, deprotection of the *N*-Boc group and HPLC purification gave compound **30** in 6% yield over a three-step pathway similar to that shown in Scheme 2 and Scheme 3 (Scheme 5). In the synthesis of MKN-1 derivatives with a sulfidyl group, the alcohols bearing *tert*-butyl sulfidyl, *iso*-propyl sulfidyl and benzenesulfidyl groups (**31–33**) were prepared in 77%, 70% and 88% yields, respectively, by treatment of compound **19** [70] with sodium *tert*-butylthiolate, sodium 2-propanethiolate or sodium thiophenolate. Then, condensation of the hydrolysate of compound **15** with alcohols **31**–**33**, followed by deprotection of the *N*-Boc group and HPLC purification gave compounds **34**–**36** in 12%, 19% and 9% yields, respectively, in three steps similar to those described above in Scheme 5. In the synthesis of an MKN-1 derivative with a methanesulfonyl group, the condensation of the hydrolysate of compound **15** with the alcohol (**20**) led to the ester (**37**) in 35% yield in two steps, and subsequent oxidation of the sulfidyl group of **37** with *m*-chloroperoxybenzoic acid (*m*CPBA), deprotection of the *N*-Boc group and HPLC purification gave compound **38** in 8% yield over two steps (Scheme 5).

### 3.3. Evaluation of Anti-HIV Activity and Cytotoxicity of the Synthesized Compounds

The anti-HIV activity of the synthesized compounds was assessed based on protection against HIV-1 (NL4-3 strain)-induced cytopathogenicity in MT-4 cells by an MTT assay [27,28,29,30]. The cytotoxicity of these compounds was determined based on reduction of the viability of MT-4 cells determined by an MTT assay. These results are shown in Table 1.

MKN-1A (**1A**), which is exactly MKN-1 (**1**), showed significant anti-HIV activity, and its diastereoisomer MKN-1B (**1B**) showed moderate anti-HIV activity, suggesting that chiral recognition by a target molecule, possibly a CA protein, might be important. These compounds were also evaluated by a different method, using an enzyme-linked immune sorbent assay (ELISA) based on their inhibitory effect against the viral p24 antigen expression in NL4-3 strain-infected MT-4 cells. The results agree with the MTT assay in that MKN-1A (**1A**) showed higher anti-HIV activity than MKN-1B (**1B**). MKN-1 (**1**), which was stereoselectively synthesized and corresponds to MKN-1A (**1A**), showed high anti-HIV activity. The cytotoxicities of MKN-1 (**1**) (MKN-1A (**1A**)) and MKN-1B (**1B**) at essentially the same level were moderate and weak. MKN-1 derivatives with 1-naphthyl, 2-naphthyl, benzofuranyl and benzothiophenyl groups (**22**, **24**, **27**, and **28**) showed weak anti-HIV activity, indicating that the indolyl group is critical and cannot be modified. These MKN-1 derivatives (**22**, **24**, **27**, and **28**) showed relatively weak cytotoxicity similar to that of MKN-1 (**1**), MKN-1A (**1A**) and MKN-1B (**1B**). An MKN-1 derivative with a methoxy group in place of the methanesulfidyl group (**30**) showed weak anti-HIV activity, suggesting that a sulfur atom is important for significant anti-HIV activity, although an oxygen atom in this position retains minor activity. Compound **30** however failed to exhibit any cytotoxicity below 50 μM, indicating that an oxygen atom is more suitable than a sulfur atom in terms of low cytotoxicity. MKN-1 derivatives with *tert*-butyl sulfidyl, *iso*-propyl sulfidyl and benzenesulfidyl groups (**34**–**36**) showed higher anti-HIV activity than the 1-naphthyl, 2-naphthyl, benzofuranyl and benzothiophenyl group-substituted derivatives (**22**, **24**, **27**, and **28**) but lower activity than the parent compound MKN-1 (**1**) (MKN-1A (**1A**)). These derivatives (**34**–**36**) exhibited relatively high cytotoxicity, compared with other derivatives. This suggests that sulfidyl groups are critical for significant anti-HIV activity but sulfidyl groups which are bulky are not suitable in terms of cytotoxicity. A sulfone-substituted derivative (**38**) showed weak anti-HIV activity and failed to exhibit any cytotoxicity below 50 μM, indicating that sulfidyl but not sulfonyl groups are critical for significant anti-HIV activity.

## 4. Discussion

Taken together, two pharmacophore functional groups (indolyl and sulfidyl groups) of MKN-1 (**1**), which was originally designed as a dipeptide mimic of Trp184 and Met185, are both important for high anti-HIV activity and should not be modified. The indolyl moiety cannot be changed into a 1-naphthyl, 2-naphthyl, benzofuranyl or benzothiophenyl group. The methanesulfidyl moiety can be converted into other sulfidyl groups, such as *tert*-butyl sulfidyl, *iso*-propyl sulfidyl and benzenesulfidyl with a slight decrease of anti-HIV activity, and into a methoxy group with a significant decrease of anti-HIV activity, but when changed into a methanesulfonyl group leads to total loss of activity. The Trp184 and Met185 residues are extremely conserved among the proteins of various HIV-1 subtypes circulating in nature, possessing the Shannon entropy scores of 0.0012 and 0.0014 for W184 and M185, respectively, which are even much lower values as compared with those of highly conserved active sites of HIV-1 integrate [71]. The data indicate exceedingly strong selective constraints against changes in the Trp184-Met185 dipeptides and suggest potential therapeutic benefits in targeting them to reduce the risk of emergence of drug resistance variants. As described in Section 3.1.1, viral mutants with Trp184Ala and Met185Ala mutations has no infectivity, causing abnormal morphology of the viral particles [64]. Therefore, the dipeptide mimic might have advantages when it is used with the lead compounds targeting capsid proteins, such as GS-6207, having different site of actions [57]. Such combinational use of compounds could increase antiviral effects via synergistic effects for disturbing capsid assembly/disassembly during HIV-1 replication in the cells and would reduce risk of emergence of drug resistance variants more significantly than the single-compound use would. However, to obtain more potent lead compounds based on the present results, pharmacophore models on the target site, i.e., the CA-CA interface, can be constructed via molecular dynamics study. The approach may be beneficial to perform alternative screening, de novo design and optimization of lead compounds for obtaining compounds with high antiviral activity. In addition, de novo design of the candidates can be performed by using structural information of a few amino acid residues flanking the W184M185 residues of the CA protein dimer. This approach may improve the binding affinity.

## 5. Conclusions

In conclusion, this study presents a new class of small molecules, which was designed by in silico screening as a dipeptide mimic of Trp184 and Met185 at the hydrophobic interaction site between two CA molecules that have been reported to be important for stabilization of the multimeric structure of CA. The designed compound MKN-1 (**1**) has significant anti-HIV-1 activity, and its diastereoisomer MKN-1B (**1B**) has lower activity. Structure activity relationship (SAR) studies of MKN-1 derivatives reveal the importance of two pharmacophore groups: indolyl and sulfidyl. In this study, whether MKN-1 (**1**) actually binds to the CA protein was not confirmed, but the chiral recognition of its target molecule was confirmed according to the difference of potencies between MKN-1 (**1**) and its diastereoisomer. The present results should be useful in the future design of a novel class of anti-HIV agents.

## Data Availability

The data presented in this study are available as Appendix A. These include the details of synthesis and characterization data of the compounds (^1^H, ^13^C-NMR, [α]_D_, IR, HRMS) including analytical HPLC charts to check the purity of the compounds.

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
