# Peer review of "Small-Molecule Anti-HIV-1 Agents Based on HIV-1 Capsid Proteins"

_biomolecules, 2021, doi:10.3390/biom11020208_

Round 1
Reviewer 1 Report
Kobayakawa et al. have presented a short communication on a set of potential anti-HIV-1 antiretrovirals that target capsid oligomerization.
Overall, the article is generally well presented however the issues below should be addressed.
Introduction:
- The authors should improve the sentences on lines 33/34, 56/57.
Results/Methods:
- The docking studies are intriguing and creative. However, the methods section for docking studies is sparse. The authors should state which exact chains from the structure they used to ‘dock and fix” as receptor/ligand as the conformations vary between monomers. As well, what were the parameters used for screening for the linker database? Was there any minimalization done for the W184M185-Capsid structure? What about for the docked identified ligands? Was a separate docking experiment done for the identified ligands, if so what parameters were used?
- Some results from the raw linker library would be beneficial. How many ligands were identified? Could these be provided in the supplement? More importantly, for the parameters used to select ligands (binding affinity, ligand efficiency, TPSA, SLogP), could this be shown in a table for each identified “hit” ligand, or at least MKN-1 (only some of these measures are provided). SlogP is stated to be a property screened for in the results but not the methods.
- A figure overlaying or showing side-by-side, the conformations of the W184M185 dimer and MKN-1 bound to capsid would compare the predicted binding of the two inhibitors.
- It is confusing and unclear generally the differences between MKN-1, -1A and -1B. As well in Figure 2, where MKN-1A/B is annotated, explicitly stating the chirality of the two forms would be help understanding, or even showing both structures side by side. Instead MKN-1A is currently labelled as 1A and 1B.
- If MKN-1A is no different from MKN-1, why are the results different between the two forms for the MTT assay?
- How were MKN-1A and -1B purified from one another for the MTT assay? This was unclear to me. I did not find this in the methods.
- The methods state that the MTT and ELISA experiments are reported as a mean of at least 2 experiments. Does this vary – i.e. do some have more than two, which and why? More explicit N values would be a benefit. Further, no standard deviation or error is reported throughout which should be added. Statistical tests would also be beneficial as we are unable to see the variability in these assays. Finally, were these experiments done as a single replicate or in duplicate, triplicate or quadruplicate? This should be made explicit in the methods.
- More information on how EC50 was calculated would be useful.
- For the EC50 experiments, was a drug carrier used? If so, was the carrier kept constant across treatments? Was there a carrier control?
- The methods should report in detail the sources for all cells and reagents, including the AZT and AMD3100 molecules. Methodological details on how cells were maintained and cultured are necessary.
- HPLC plots would be benefited with numbering of the axes.
- The compounds are added as TFA salts, but TFA can be toxic to the cells – could this be having an effect on viability assays? Could the salt be exchanged some way to the acetate or hydrochloride forms with reverse-phase HPLC or removed entirely?
Author Response
Correspondences to Reviewer 1:
Thank you for the reviewer’s positive comments.
Introduction:
- The authors should improve the sentences on lines 33/34, 56/57.
Thank you for the comments. According to the reviewer’s suggestion,
33/34 “Human immunodeficiency virus type 1 (HIV-1) is a member of a retrovirus family, which infects CD4-positive T-cells or macrophage, eventually causing AIDS.”
was changed into:
“As a retrovirus, human immunodeficiency virus type 1 (HIV-1) can infect CD4-positive T-cells or macrophage, eventually causing AIDS.”
56/57 “However, if small compounds with inhibitory activity against viral uncoating and assembly are found, these compounds might themselves have cell membrane permeability.”
was changed into:
“However, small compounds that are found to have inhibitory activity against viral uncoating and assembly might have cell membrane permeability.”
Results/Methods:
- The docking studies are intriguing and creative. However, the methods section for docking studies is sparse. The authors should state which exact chains from the structure they used to ‘dock and fix” as receptor/ligand as the conformations vary between monomers. As well, what were the parameters used for screening for the linker database? Was there any minimalization done for the W184M185-Capsid structure? What about for the docked identified ligands? Was a separate docking experiment done for the identified ligands, if so what parameters were used?
Thank you for the comments. According to the reviewer’s suggestion, we have added more information on the method of our in silico screening in the “Supplementary Materials” (blue portions).
In silico screening of antiviral candidates
To perform the in silico screening, we first obtained the structure of the dimer of CA proteins (PDB ID:3J34) from the Protein Data Bank (https://www.rcsb.org/). The structure of the dimer of CA proteins was thermodynamically optimized by the energy minimization using MOE and the Amber10:EHT force field.S5,S6 The one monomer of the dimer of CA proteins was fixed as a receptor, whereas the residues of the other monomer were removed except for Typ184 and Met185 residues, of which the side chains play a key role of the dimer formation. Using the complex composing of the CA monomer as a receptor and Trp184Met185 dipeptide as a ligand, we searched for the compounds having a higher affinity than the dimer formation. To do this, the main chain backbone replacement of the dipeptide on the CA protein was performed by the Scaffold Replacement application in MOE using the linker database of MOE and the Amber10:EHT force field, whereas the side chains of the dipeptide were fixed. From the result, we selected the compounds having higher scores of the binding affinity for receptor (London dG), ligand efficacy, and topological polar surface area.
[S5] Case, D.A.; Darden, T. A.; Cheatham, T. E. et al. AMBER 10, University of California, San Francisco 2008.
[S6] Gerber, P.R.; Müller, K. MAB, a generally applicable molecular force field for structure modelling in medicinal chemistry. J. Computer-Aided Mol. Des. 1995, 9, 251–268.
- Some results from the raw linker library would be beneficial. How many ligands were identified? Could these be provided in the supplement? More importantly, for the parameters used to select ligands (binding affinity, ligand efficiency, TPSA, SLogP), could this be shown in a table for each identified “hit” ligand, or at least MKN-1 (only some of these measures are provided). SlogP is stated to be a property screened for in the results but not the methods.
Thank you for the interest. We have obtained 4 candidates by the present approach. Since we plan to submit a manuscript with these molecules, we would like to avoid presenting the information related to them. Instead, we have added more information on the parameters for the MKN-1 selection.
line 104-105: After “…MKN-1 (1), whose London dG value,” “ligand efficiency, topological polar surface area (TPSA) and SlogP value are -9.134 kcal/mol, 0.2559, 69.73 Å2 and 4.022, respectively” was added.
- A figure overlaying or showing side-by-side, the conformations of the W184M185 dimer and MKN-1 bound to capsid would compare the predicted binding of the two inhibitors.
Thank you for the suggestion. According to this comment, we made a new figure 2A showing side-by-side, the conformations of the W184M185 dimer and MKN-1 bound to capsid.
- It is confusing and unclear generally the differences between MKN-1, -1A and -1B. As well in Figure 2, where MKN-1A/B is annotated, explicitly stating the chirality of the two forms would be help understanding, or even showing both structures side by side. Instead MKN-1A is currently labelled as 1A and 1B.
Thank you for the suggestion. According to this comment, absolute configurations of MKN-1A and MKN-1B were drawn in Scheme 2, which were determined in the next section (3.2.2. Stereoselective synthesis of MKN-1 (1)). In addition, in Scheme 3 “MKN-1 (1) = MKN-1A (1A))” was shown.
- If MKN-1A is no different from MKN-1, why are the results different between the two forms for the MTT assay?
Thank you for the suggestion. Concerning this comment, values obtained in cell-based anti-HIV assays are usually flexible due to conditions of virus and cells. The authors think this difference might be inside scope validation between assays.
- How were MKN-1A and -1B purified from one another for the MTT assay? This was unclear to me. I did not find this in the methods.
Thank you for the suggestion. Concerning this comment, in Scheme 2 HPLC purification was shown about separation of MKN-1A and MKN-1B. In addition, in the “Supplementary Materials” page 7-8, the experimental procedure is described.
- The methods state that the MTT and ELISA experiments are reported as a mean of at least 2 experiments. Does this vary – i.e. do some have more than two, which and why? More explicit N values would be a benefit. Further, no standard deviation or error is reported throughout which should be added. Statistical tests would also be beneficial as we are unable to see the variability in these assays. Finally, were these experiments done as a single replicate or in duplicate, triplicate or quadruplicate? This should be made explicit in the methods.
Thank you for the suggestion. Concerning this comment, sequential diluted concentrations of compounds are used. Thus, at least two experiments might be enough. But, according to this comments, in Table 1 the numbers of experiments for each compound are shown inside parentheses, and the data are shown as a mean value or mean value ± a standard deviation in case of more than three experiments.
- More information on how EC50 was calculated would be useful.
Thank you for the suggestion. Concerning this comment, more information on how EC50 was calculated is described in the Table 1 legend and the “Supplementary Materials” page 30-31.
- For the EC50 experiments, was a drug carrier used? If so, was the carrier kept constant across treatments? Was there a carrier control?
Thank you for the suggestion. Concerning this comment, a drug carrier was not used.
- The methods should report in detail the sources for all cells and reagents, including the AZT and AMD3100 molecules. Methodological details on how cells were maintained and cultured are necessary.
Thank you for the suggestion. According to this comment, the sources for all cells and reagents, including the AZT and AMD3100 molecules, and methodological details on how cells were maintained and cultured were described in the “Supplementary Materials” page 30-31. Reference S8 was added. 
[S8] Harada, S.; Koyanagi, Y.; Yamamoto, N. Infection of HTLV-III/LAV in HTLV-I-carrying cells MT-2 and MT-4 and application in a plaque assay. Science 1985, 229, 563-566.
- HPLC plots would be benefited with numbering of the axes.
Thank you for the suggestion. According to this comment, HPLC charts were shown with numbering of the axes in the “Supplementary Materials”.
- The compounds are added as TFA salts, but TFA can be toxic to the cells – could this be having an effect on viability assays? Could the salt be exchanged some way to the acetate or hydrochloride forms with reverse-phase HPLC or removed entirely?
Thank you for the suggestion. Concerning this comment, some compounds with TFA salts did not show cytotoxicity until 50 micro M. Thus, the authors think the use of TFA salts in cells is OK.

Reviewer 2 Report
The authors describe the design of some organic compounds for inhibiting assembly of the human immunodeficiency virus (HIV) capsid. They provide preliminary evidence for moderate inhibition of HIV infection of cultured cells by some of the designed compounds. The experiments appear to have been competently done. The results may provide a starting point for further design of potential anti-HIV compounds.
The manuscript does not include citations to most of the many published articles on a substantial number of HIV capsid assembly-inhibiting compounds. This deficiency should be addressed.
Author Response
Correspondences to Reviewer 2:
Thank you for the reviewer’s positive comments.
- The manuscript does not include citations to most of the many published articles on a substantial number of HIV capsid assembly-inhibiting compounds. This deficiency should be addressed.
Thank you for the suggestion. According to this comment, the number of reference 16 was increased: 5 papers. Thus, original “16k” was changed into “16o”.

Reviewer 3 Report
The manuscript by Kobayakawa and colleagues reports the design and synthesis of small molecules targeting HIV-1 capsid. The target is novel and potential applications are exciting and much needed. The problem with the study is that it is secondary to a paper by Link et al. in Nature Medicine and Nature, which reported on a small molecule targeting capsid with picomolar EC50. In contrast, the compound described in this study has EC50 above 10 uM. It is unclear whether the potency can be increased by chemical modifications, and the manuscript does not provide any clues in this regard. It appears that the selected site on the CA may not be optimal for inhibition, or that the virtual screen used for selection of the candidates should be optimized.
Author Response
Correspondences to Reviewer 3:
Thank you for the reviewer’s positive comments.
- It is unclear whether the potency can be increased by chemical modifications, and the manuscript does not provide any clues in this regard. It appears that the selected site on the CA may not be optimal for inhibition, or that the virtual screen used for selection of the candidates should be optimized.
Thank you for the suggestion. According to this comment, to clarify our plan to obtain more potent lead compounds, the following sentences was added in “Discussion.”
“To obtain more potent lead compounds based on present results, however, pharmacophore models on the target site, i.e., the CA-CA interface, can be constructed via molecular dynamics study. The approach may be beneficial to perform alternative screening, de novo design, and optimization of lead compounds for obtaining compounds with high antiviral activity. In addition, de novo design of the candidates can be performed by using structural information of a few amino acid residues flanking the W184M185 residues of the CA protein dimer. This approach may improve the binding affinity.”

Round 2
Reviewer 3 Report
The authors' response does not address my concern. I still do not see any benefit of targeting the dipeptide selected by the authors when a 1000-fold higher activity was reported by Link et al. using a different target. This should be acknowledged, and a discussion should be included to explain these differences.
Author Response
Correspondences to Reviewer 3:
The authors' response does not address my concern. I still do not see any benefit of targeting the dipeptide selected by the authors when a 1000-fold higher activity was reported by Link et al. using a different target. This should be acknowledged, and a discussion should be included to explain these differences.
Thank you for your useful comment. According to this comment, the following sentences with ref [24] were added in Discussion.
The Trp184 and Met185 residues are extremely conserved among the proteins of various HIV-1 subtypes circulating in nature, possessing the Shannon entropy scores of 0.0012 and 0.0014 for W184 and M185, respectively, which are even much lower values as compared with those of highly conserved active sites of HIV-1 integrate [24]. The data indicate exceedingly strong selective constraints against changes in the Trp184-Met185 dipeptides and suggest potential therapeutic benefits in targeting them to reduce the risk of emergence of drug resistance variants. As described in 3.1.1. structural analysis of CA proteins, viral mutants with Trp184Ala and Met185Ala mutations has no infectivity, causing abnormal morphology of the viral particles [18]. Therefore, the dipeptide mimic might have desirable advantages as an alternative candidate of reported lead compounds such as GS-6207 [16o].
[24] Takahata, T.; Takeda, E.; Tobiume, M. et al. Critical contribution of Tyr15 in the HIV-1 integrase (IN) in facilitating IN assembly and nonenzymatic function through the IN precursor form with reverse transcriptase. J. Virol. 2016, 91, e02003-16.
Round 3
Reviewer 3 Report
I think the manuscript would benefit from a better justification of the need for an alternative target on capsid. What benefit would be achieved by using two compounds targeting different sites on p24?
Author Response
Correspondences to Reviewer 3:
I think the manuscript would benefit from a better justification of the need for an alternative target on capsid. What benefit would be achieved by using two compounds targeting different sites on p24?
Thank you for your useful comment. According to this comment, the sentences in the Discussion were changed.
“Therefore, the dipeptide mimic might have desirable advantages as an alternative candidate of reported lead compounds such as GS-6207 [16o].”
was changed into
“Therefore, the dipeptide mimic might have advantages when it is used with the lead compounds targeting capsid proteins, such as GS-6207, having different site of actions [16o]. Such combinational use of compounds could increase antiviral effects via synergistic effects for disturbing capsid assembly/disassembly during HIV-1 replication in the cells and would reduce risk of emergence of drug resistance variants more significantly than the single-compound use would.”